# Tuning the Biological Activity of PI3K*δ* Inhibitor by the Introduction of a Fluorine Atom Using the Computational Workflow

**DOI:** 10.3390/molecules28083531

**Published:** 2023-04-17

**Authors:** Wojciech Pietruś, Mariola Stypik, Marcin Zagozda, Martyna Banach, Lidia Gurba-Bryśkiewicz, Wioleta Maruszak, Arkadiusz Leniak, Rafał Kurczab, Zbigniew Ochal, Krzysztof Dubiel, Maciej Wieczorek

**Affiliations:** 1Department of Medicinal Chemistry, Maj Institute of Pharmacology, Polish Academy of Sciences, Smetna 12, 31-343 Krakow, Poland; 2Celon Pharma S.A., ul. Marymoncka 15, 05-152 Kazuń Nowy, Poland; 3Faculty of Chemistry, Warsaw University of Technology, ul. Nowakowskiego 3, 00-664 Warsaw, Poland

**Keywords:** PI3K*δ*, asthma, CPL302415, induced-fit docking, QPLD, MD, fluorine, MM-GBSA, molecular docking

## Abstract

As a member of the class I PI3K family, phosphoinositide 3-kinase *δ* (PI3K*δ*) is an important signaling biomolecule that controls immune cell differentiation, proliferation, migration, and survival. It also represents a potential and promising therapeutic approach for the management of numerous inflammatory and autoimmune diseases. We designed and assessed the biological activity of new fluorinated analogues of CPL302415, taking into account the therapeutic potential of our selective PI3K inhibitor and fluorine introduction as one of the most frequently used modifications of a lead compound to further improve its biological activity. In this paper, we compare and evaluate the accuracy of our previously described and validated in silico workflow with that of the standard (rigid) molecular docking approach. The findings demonstrated that a properly fitted catalytic (binding) pocket for our chemical cores at the induced-fit docking (IFD) and molecular dynamics (MD) stages, along with QM-derived atomic charges, can be used for activity prediction to better distinguish between active and inactive molecules. Moreover, the standard approach seems to be insufficient to score the halogenated derivatives due to the fixed atomic charges, which do not consider the response and indictive effects caused by fluorine. The proposed computational workflow provides a computational tool for the rational design of novel halogenated drugs.

## 1. Introduction

The inhibition of phosphoinositide 3-kinase (PI3K), especially the first class of this family of lipid kinases consisting of *α*, *β*, *γ*, and *δ* subunits, is a promising approach for the treatment of many inflammatory and autoimmune diseases, such as systemic lupus erythematosus or multiple sclerosis [1,2,3]. Because PI3K is involved in many cellular processes, including proliferation, growth, migration, metabolism regulation, and embryogenesis (characterized by high expression of this protein in different human cells), it is considered an excellent therapeutic target [4,5].

Fluorine, which is slightly larger than hydrogen and highly electronegative, constitutes a remarkable role in medicinal chemistry [6,7,8,9,10,11]. Currently, fluorination is a standard strategy to improve the bioavailability of designed drugs and optimize their biological activity [12]. In the last decade, nearly 30% of the drugs approved by the US Food and Drug Administration (FDA) contained fluorine, and fluorinated pharmaceuticals accounted for over 50% of the most profitable drugs worldwide [6,13].

Molecular modeling methods, such as molecular docking, molecular dynamics (MD), or free-energy perturbation, are widely applied during the rational designing of new compounds to evaluate the formation of ligand–receptor complexes [14]. However, the standard (rigid) docking method has two main limitations: (i) the conformation space is reduced due to the limitations imposed on the system (rigid receptor) and (ii) conventional scoring functions do not consider inductive or resonance effects (which is extremely important for fluorine derivatives). Nevertheless, this method can be successfully used to quickly score poses and find promising hits from a large library of compounds [15,16]. Recently, we proposed a method that can overcome some limitations of the standard molecular docking approach [7,9]. We demonstrated that a workflow comprising a combination of more sophisticated methods, such as induced-fit docking (IFD), quantum polarized ligand docking (QPLD), and binding-free energy calculations based on the Generalized Born Surface Area (GBSA), is more accurate in the prediction of the ligand–receptor complex and its energy than the standard docking procedure, but it is more computationally expensive [17].

In this work, we used the previously described in silico workflow [7,9] to design and determine the biological activity of new fluorinated analogs of CPL302415 (Figure 1), a PI3K*δ* inhibitor. The usefulness of this workflow was validated by correlating the biological activity (IC_50_) with energy changes calculated in the GBSA model.

## 2. Results and Discussion

Molecular docking of CPL302415 derivatives was performed on the previously described PI3K*δ* crystal structure (PDB ID: 2WXL). The binding mode observed for CPL302415 (**2**) was consistent with the previously reported ones [18,19] (Figure 2). The nitrogen of the (difluoromethyl)-1*H*-benzimidazole fragment interacted with the positively charged Lys779, while the morpholine ring at position 7 (which is required for interaction with PI3K*δ* at its catalytic site) formed a hydrogen bond with the main chain of Val828. The *tert*-butyl piperazine moiety bound to the Trp760 shelf through C-H⋯π/cation⋯π interaction (Figure 2).

According to molecular docking, the static nature of biomolecules is the main source of their limitations, as it does not consider the dynamic nature of biological structures [20]. The analysis of the structure–activity relationship (SAR) of our library revealed that the change from the difluoro group (**2**) to the methyl group (**1**) caused a decrease in the activity of the compound, but the introduction of the trifluoromethyl group (**3**) almost inactivated the compound (Table 1). Interestingly, the docking scores of the above-mentioned derivatives indicated that compound **1** should have the highest IC_50_ value (Table 1) in the series—even if we took into account the average docking score for the top three poses, the tendency was almost identical. We substituted position 3 with chlorine (**4**) or bromine (**5**) and the resulting compounds were the most potent based on docking scores (Table 1), but they had worse IC_50_ values than compound **2**.

To assess the accuracy of the presented method, correlation coefficients between the IC_50_ values and the obtained docking scores were calculated (for the first pose and the average of the first three poses, respectively). The tests showed that there was no correlation (0.53; *p* > 0.05, and 0.51; *p* > 0.05). 

Based on the binding mode of CPL302415 (**2**), we decided to substitute *tert*-butyl moiety with a benzyl fragment to increase the number of π–π or hydrophobic interactions with Thr750. Therefore, a series of fluorine derivatives were synthesized using the 7-(morpholin-4-yl)pyrazolo[1,5-*a*]pyrimidin-5-yl]-2-(difluoromethyl)-1*H*-benzimidazole core (**6**–**9**) as well as with an additional carbonyl group in the 1-[2-(4-benzylpiperazine-1-carbonyl) fragment (substituent in position 2 of the pyrazolo[1,5-*a*]pirymidine core) (**10**–**13**) (Table 2). We docked both nonfluorinated (in the aromatic ring) cores with benzyl fragment compounds to the 2WXL crystal structure, and the observed binding modes were similar to those of CPL302415. However, the hydrogen bonds formed by the morpholine fragment in compound **10** with Val828 were characterized by the worst geometric parameters in comparison to compound **6** (Figure 3), which could be associated with the lower activity of compound **10** (Table 2).

The carbonyl group affected the orientation of the benzylpiperazine fragment, causing a change in the cation⋯π interaction with Trp760 (Figure 3). The analysis of the IC_50_ values did not allow us to draw any conclusions because in the first series of derivatives (**6**–**9**), the *meta*-fluoro derivative **8** had the highest IC_50_ value, while in the second series (**10**–**13**), the *meta*-fluoro derivative **12** was the best (Table 2), which suggested that carbonyl oxygen had an impact on binding. Interestingly, we found a poor correlation between any docking score (nor the first best and the best three poses) and biological activity (IC_50_) (correlation coefficients were −0.02 and 0.03, respectively).

Due to the low correlation of the docking scores obtained in the standard molecular docking approach (Table 1 and Table 2) with biological activity, a previously described and validated computational workflow [7,9] was used to compensate for the limitations of conventional scoring functions and increase the accuracy of the pose prediction, especially for fluorinated derivatives (Figure 4). Because fluorine is the most electronegative element, its substitution leads to significant changes in the distribution of electron density and as a consequence, resonance and inductive effects. The standard molecular docking approach does not consider these effects, while the QPLD method uses atomic charges of a ligand calculated using the quantum mechanical (QM)/molecular mechanical (MM) approach in the protein environment for docking (Figure 4). To minimize the uncertainty of predicted poses, the binding-free energy was calculated for three poses obtained at the QPLD stage with the smallest root square mean deviation (RMSD) to the core. The energy change (ΔΔG) after fluorination was estimated in comparison to the nonfluorinated compound (Figure 4). Additionally, by using the MD of a nonfluorinated compound, our approach enabled us to relax the binding pocket and fit it into the particular core. Due to its small size [6,21], we assumed that the introduction of a fluorine atom would not result in large conformational changes.

Analysis of the MD trajectories showed that the previously described binding mode [19] and the formed interactions were highly stable (Figure 5). We found that Lys779 involved in a hydrogen bond with benzimidazole nitrogen and tryptophane, which interacted with the positively charged nitrogen of the piperazine group, had high stability and a permanent position. Additionally, Tyr813 was engaged in a weak hydrogen bond with the CH donor (Figure 5). The substitution of the *tert*-butyl fragment with the benzyl fragment allowed an additional C–H⋯π/hydrophobic interaction with Thr750. 

The comparison of the first top pose for each compound in the series obtained by applying the standard approach showed that the 7-(morpholin-4-yl)pyrazolo[1,5-*a*]pyrimidin-5-yl]-2-(difluoromethyl)-1*H*-benzimidazole core had less mobility, whereas the *tert*-butyl 1-{2-[(4-*tert*-butylpiperazin-1-yl) and 1-[2-(4-benzylpiperazine-1-carbonyl) fragment was directed toward the solvent space, and thus had higher flexibility (Figure 6I). Since fluorine is a bioisostere of hydrogen [6,8], the substitution of this element should not lead to drastic changes in the binding mode. The comparison of the observed binding modes showed that there was no coherence (Figure 6I) and that small molecular changes highly affected the position of adjacent fragments of molecules.

The results obtained using different methods of atomic charge assignment are clearly different (Figure 6I,II). Compared to standard docking, in which the *tert*-butyl piperazine or benzylpiperazine fragment resulted in high flexibility and mobility (Figure 6I), the binding poses obtained by the proposed workflow (where the atomic charges are QM-derived) showed lower RMSD (less than 0.5 Å; Figure 6II). In the case of the QPLD-docked poses, the introduced fluorine atom did not induce any drastic conformational changes (Figure 6II). However, due to its high electronegativity, this element may affect the tuning of atomic charges, pK_a_, and the basicity or acidity (electron density distribution in general) of neighboring functional groups, rather than intermolecular interactions with the biological target [6]. 

Using the MM-GBSA approach, the interaction energy of the obtained ligand–receptor complex was calculated as the average of the top three ligand poses ($\overset{-}{\Delta G}$) (Table 3 and Table 4). The correlation coefficient of the biological activity (IC_50_) and the MM-GBSA scores of the *tert*-butyl piperazine derivatives was significantly higher with the use of our approach compared to standard docking scores (0.95; *p* < 0.05 and ~0.5, respectively) (Table 3). The results obtained for the *tert*-butyl piperazine derivatives showed that the introduction of a difluoro group to the 2-methyl-1*H*-benzimidazole fragment increased the stabilization energy of the complex (ΔΔG = −7.6), whereas a trifluoro group decreased the stability of complexes, which was probably due to steric hindrances and inductive effects (Table 3). Moreover, as shown by Kurczab et al. [22], the MM-GBSA approach can be successfully used for heavier halogens. Therefore, we extended our library with derivatives containing bromine or chlorine at position 3. These derivatives were less active, but the proposed algorithm accurately predicted the energy loss for the difluoromethyl-1*H*-benzimidazole derivative (Table 3).

For 4-benzylpiperazine and 4-benzylpiperazine-1-carbonyl fluorine derivatives, the correlation coefficient was almost equal to 1 (0.9; *p* < 0.05 and 0.95; *p* < 0.05, respectively), which implies that although there was no correlation with the docking scores, we obtained a perfect correlation here (Table 4). 

In addition, to demonstrate the superiority of the proposed workflow, the QPLD docking protocol was performed on every stage of kinase flexibility: (i) on rigid crystalized PI3K*δ* structure (PDB ID: 2WXL) and (ii) on the induced-fit docked poses chosen for the MD stage. The molecular dynamics were the most time-consuming step; therefore, if, thanks to QM-derived atomic charges (which could help obtain a more accurate ligand-receptor interaction energy), a higher correlation of IC_50_ values with ΔG values in the previous stages could be achieved, the computational time could be saved compared to the presented workflow. In the same way as in the presented workflow, using the MM-GBSA approach, the interaction energy of the obtained ligand–receptor complex was calculated as the average of the top three ligand poses ($\overset{-}{\Delta G}$) (SI Appendix A). For the rigid conformation of the catalytic pocket, the correlation coefficients of the biological activity (IC_50_) and the MM-GBSA score of *tert*-butyl piperazine derivatives were slightly lower (−0.34; *p* > 0.05) compared to the standard docking scores (~0.5) and proposed workflow (0.95) (SI Appendix A). The correlation coefficients for 4-benzylpiperazine (−0.43; *p* > 0.05) and 4-benzylpiperazine-1-carbonyl fluorine derivatives (−0.78; *p* > 0.05) were better than this obtained with the standard (rigid) approach (~0.0) but lower than those obtained with our workflow (~0.93). Next, the QPLD approach was used on grids obtained in the IFD protocol for compounds **1**, **6**, and **10**. The correlation coefficient for *tert*-butyl piperazine derivatives was similar (−0.35; *p* > 0.05) to that obtained with a rigid catalytic pocket (−0.34), whereas the correlation was better for 4-benzylpiperazine (0.60; *p* > 0.05) and 4-benzylpiperazine-1-carbonyl fluorine derivatives (−0.78; *p* > 0.05) compared to those obtained with the non-flexible structure of PI3K*δ* (−0.43; −0.78, respectively). Nevertheless, the correlation coefficient obtained with the described workflow was higher (~0.93), which confirms the importance of all stages in the proposed workflow.

It is worth stressing that the correlation of IC_50_ values with ΔG values increases with greater flexibility of the catalytic pocket and its fit to the molecular core. These results suggest that properly prepared catalytic pockets for chemical cores in the IFD and MD stages, combined with QM-derived atomic charges, can be effectively used for prediction as well as to improve the discrimination of active compounds from inactive ones.

## 3. Materials and Methods

### 3.1. Chemistry

The compounds **1**–**14** discussed in this work were synthesized following the general procedures presented in our previous papers [19,23]. The synthesis pathway has multiple steps, of which the last one is described here. The final compounds were synthesized via reactions such as reductive amination, amidation, or coupling reactions including the Buchwald–Hartwig reaction, and their yields varied depending on the structure. The Buchwald–Hartwig reaction, amidation reaction, and reductive amination are especially described in this work.

### 3.2. General Information

The reagents (at least 95% purity) were purchased from ABCR (Dallas, TX, USA), Acros (Geel, Belgium), Alfa Aesar (Haverhill, MA, USA), Combi-Blocks (San Diego, CA, USA), Fluorochem (Hadfield, UK), (Buchs, Switzerland), Merck (Darmstadt, Germany), and Sigma Aldrich (Saint Louis, MI, USA) and were used without additional purification. Solvents were purified according to the standard procedures if required. Air- or moisture-sensitive reactions were carried out under an argon atmosphere. The progress of all reactions was routinely monitored by thin-layer chromatography (TLC). TLC was performed on silica gel-coated plates (Kieselgel F254), which were visualized using a UV light. Flash chromatography was performed on Merck silica gel 60 (230–400 mesh ASTM). ^1^H NMR spectra were acquired using JOEL JNMR-ECZS 400 and 600 MHz spectrometers (^1^H observed at 400, and 600 MHz, respectively). ^13^C NMR spectra were recorded at 101 and 151 MHz, respectively. Due to the poor solubility of some final compounds, the usual characterization using ^13^C NMR was omitted. Chemical shifts for ^1^H and ^13^C NMR spectra were reported in *d* (ppm) using tetramethylsilane as an internal standard or based on the residual undeuterated solvent signal (2.50 ppm for DMSO-*d*_6_ and 7.26 ppm for CDCl_3_). The abbreviations for multiplets of ^1^H signals are as follows: s (singlet), d (doublet), t (triplet), m (multiplet), dd (doublet of doublets), dt (doublet of triplet), and q (quartet). Coupling constants (J) are expressed in Hertz. A JEOL Royal HFX probe head was used for recording the ^13^C NMR spectrum as it allows measurements to be taken with the simultaneous decoupling of both ^1^H and ^19^F nuclei [24]. Atmospheric pressure ionization and electrospray ionization mass spectra were obtained using an Agilent 6130 LC/MSD spectrometer or Agilent 1290 UHPLC system coupled with an Agilent QTOF 6545 mass spectrometer. All spectra of the final compounds are shown in the Appendix A.

### 3.3. Synthesis

Compounds **1** and **2** were synthesized according to the procedures described in our previous publications [19,23].

#### 3.3.1. General Procedure for the Buchwald–Hartwig Reaction

To a pressure, microwave vessel, 5-chloro-pyrazolo[1,5-*a*]pyrimidine (1.0 eq), amine (1.5 eq), tris(dibenzylideneacetone)dipalladium (0.05 eq), 9,9-dimethyl-4,5-bis(diphenyl phosphino)xanthene (0.1 eq), cesium carbonate (2.0 eq), and solvent (10 mL/1 g pyrazolo[1,5-*a*]pyrimidine) were added. The reaction vessel was then sealed and heated to 150 °C for 6 h in a microwave oven at 200 W. After heating, the reaction mixture was filtered through Celite^®^ and concentrated, and the resulting crude product was purified by flash chromatography.

1-{2-[(4-*Tert*-butylpiperazin-1-yl)methyl]-7-(morpholin-4-yl)pyrazolo[1,5-*a*]pyrimidin-5-yl}-2-(trifluoromethyl)-1*H*-benzimidazole (**3**).Compound **3** was synthesized from 4-{2-[(4-*tert*-butylpiperazin-1-yl)methyl]-5-chloropyrazolo[1,5-*a*]pyrimidin-7-yl}morpholine (0.88 g, 2.04 mmol), 2-(trifluoromethyl)-benzimidazole (0.57 g, 3.06 mmol), tris(dibenzylideneacetone)dipalladium (93.3 mg, 0.102 mmol), 9,9-dimethyl-4,5-bis(diphenylphosphino)xanthene (124.0 mg, 0.204 mmol), cesium carbonate (1.34 g, 4.08 mmol), and toluene (8.0 mL), according to the general procedure for the Buchwald–Hartwig reaction. The resulting crude product was purified by flash chromatography (0–100% AcOEt gradient in heptane, amine-functionalized gel column) to obtain compound **3** as a white solid (38.0 mg, 0.07 mmol) with a 3% yield.^1^H NMR (600 MHz, CDCl_3_) *δ* 7.94–7.93 (m, 1H, Ar-H), 7.55–7.53 (m, 1H, Ar-H), 7.46–7.42 (m, 2H, Ar-H), 6.63 (s, 1H, Ar-H), 6.16 (s, 1H, Ar-H), 3.98–3.97 (m, 4H, morph.), 3.90–3.89 (m, 4H, morph.), 3.82 (s, 2H, CH_2_), 2.67 (d, *J* = 2.1 Hz, 8H, piperaz.), and 1.13–1.06 (m, 9H, *t*-Bu.).^13^C{1H, 19F}NMR (151 MHz, CDCl_3_) *δ* 151.2, 150.2, 147.0, 141.0, 139.9, 135.4, 126.3, 124.5, 121.7, 119.7, 118.0, 111.9, 97.2, 88.1, 66.2, 56.3, 53.8, 48.6, 45.7, 45.0, 29.7, 25.9, and 25.8.HRMS (ESI/MS): *m*/*z* calculated for C_27_H_33_F_3_N_8_O [M + H]^+^ 543.2802, found 543.2806.

1-{2-[(4-*Tert*-butylpiperazin-1-yl)methyl]-3-chloro-7-(morpholin-4-yl)pyrazolo[1,5-*a*]pyrimidin-5-yl}-2-(difluoromethyl)-1*H*-benzimidazole (**4**).To the solution of 1-{2-[(4-*tert*-butylpiperazin-1-yl)methyl]-7-(morpholin-4-yl)pyrazolo[1,5-*a*]pyrimidin-5-yl}-2-(difluoromethyl)-1*H*-benzimidazole (230.0 mg, 0.44 mol, 1.1 eq) in dichloromethane (DCM) (4 mL), N-chlorosuccinimide (64.4 mg, 0.48 mmol) was added. The reaction mixture was stirred for 1 h at room temperature. Then, sodium metabisulfite (3 mL) and water (5 mL) were added and the aqueous mixture was extracted with DCM (3 × 5 mL). The combined organic extracts were washed with water, dried over Na_2_SO_4_, filtered, and concentrated. The resulting crude product was purified by flash chromatography (0–100% ethyl acetate gradient in heptane, amine-functionalized gel column) and crystallization (AcOEt) to obtain compound **4** (134.0 mg; 0.24 mmol) as a white solid with a 55% yield. ^1^H NMR (600 MHz, CDCl_3_) *δ* 7.92–7.91 (m, 1H, Ar-H), 7.70–7.69 (m, 1H, Ar-H), 7.46–7.40 (m, 2H, Ar-H), 7.31 (t, *J* = 53.6 Hz, 1H, CHF_2_), 6.35 (s, 1H, Ar-H), 3.98–3.96 (m, 4H, morph.), 3.94–3.91 (m, 6H, morph.), 2.71–2.65 (m, 8H), and 1.10 (s, 9H, *t*-Bu.).^13^C{^1^H, ^19^F}NMR (151 MHz, CDCl_3_) *δ* 151.3, 150.4, 148.1, 145.2, 144.7, 141.9, 134.5, 125.9, 124.3, 121.6, 111.8, 109.4 (CF_2_), 87.7, 66.2, 53.3, 48.7, 45.7, and 25.9 (*t*-Bu.).HRMS (ESI/MS): *m*/*z* calculated for C_27_H_33_ClF_2_N_8_O [M + H]^+^ 558.2434, found 538.2442.

1-{3-Bromo-2-[(4-*tert*-butylpiperazin-1-yl)methyl]-7-(morpholin-4-yl)pyrazolo[1,5-*a*]pyrimidin-5-yl}-2-(difluoromethyl)-1*H*-benzimidazole (**5**).To the solution of 1-{2-[(4-*tert*-butylpiperazin-1-yl)methyl]-7-(morpholin-4-yl)pyrazolo[1,5-*a*]pyrimidin-5-yl}-2-(difluoromethyl)-1*H*-benzimidazole (500.0 mg, 0.94 mol) in DCM (7 mL), N-bromosuccinimide (204.0 mg, 1.13 mmol, and 1.2 eq) was added. The reaction mixture was stirred for 1 h at room temperature. Then, sodium metabisulfite (5 mL) and water (10 mL) were added and the aqueous mixture was extracted with DCM (3 × 10 mL). The combined organic extracts were washed with water, dried over Na_2_SO_4_, filtered, and concentrated. The resulting crude product was purified by flash chromatography (0–100% ethyl acetate gradient in heptane, amine-functionalized gel column) and crystallization (AcOEt) to obtain compound **5** (370.0 mg; 0.61 mmol) as a white solid with a 65% yield.^1^H NMR (600 MHz, CDCl_3_) *δ* 7.92–7.91 (m, 1H, Ar-H), 7.71–7.70 (m, 1H, Ar-H), 7.46–7.40 (m, 2H, Ar-H), 7.39 (t, *J* = 53.5 Hz, 1H, CHF_2_), 6.37 (s, 1H, Ar-H), 3.98–3.96 (m, 4H, morph.), 3.94–3.92 (m, 4H, morph.), 3.90 (s, 2H, CH_2_), 2.81–2.57 (m, 8H), and 1.08 (s, 9H, *t*-Bu.). ^13^C{^1^H, ^19^F}NMR (151 MHz, CDCl_3_) *δ* 151.9, 151.4, 148.4, 146.6, 144.8, 141.9, 134.5, 125.9, 124.3, 121.6, 111.8, 109.4 (CF_2_), 87.7, 66.2, 53.6, 48.7, 45.7, and 25.8 (*t*-Bu.).HRMS (ESI/MS): *m*/*z* calculated for C_27_H_33_BrF_2_N_8_O [M + H]^+^ 602.1929, found 602.1936.

#### 3.3.2. General Procedure for the Reductive Amination Reaction

To the solution of the corresponding aldehyde (1.0 eq) in a dry DCM (10 mL/1 g aldehyde), an amine derivative (1.2 eq) was added, and the reaction mixture was stirred for 1 h at room temperature. Then, sodium triacetoxyborohydride (1.5 eq) was added and the mixture was stirred for a further 15 h at room temperature. Next, water was added to the reaction mixture and the phases were separated. The aqueous phase was extracted three times with DCM, while the combined organic phases were dried over anhydrous sodium sulfate, filtered, and concentrated. The resulting residue was purified by flash chromatography.

1-{2-[(4-Benzylpiperazin-1-yl)methyl]-7-(morpholin-4-yl)pyrazolo[1,5-*a*]pyrimidin-5-yl}-2-(difluoromethyl)-1*H*-benzimidazole (**6**).Compound **6** was prepared from aldehyde 5-[2-(difluoromethyl)-1*H*-benzimidazol-1-yl]-7-(morpholin-4-yl)pyrazolo[1,5-*a*]pyrimidine-2-carbaldehyde (0.15 g, 0.37 mmol), 1-benzylpiperazine (0.80 g, 0.45 mmol) as an amine, DCM (3.0 mL), and sodium triacetoxyborohydride (0.12 g, 0.56 mmol), according to the general procedure for reductive amination reaction. The resulting crude product was purified by flash chromatography (0–10% MeOH gradient in AcOEt) to obtain compound **6** (0.1 g, 0.18 mmol) with a 47% yield. ^1^H NMR (600 MHz, DMSO-*d*_6_) *δ* 7.88–7.87 (m, 1H, Ar-H), 7.81 (d, *J* = 7.6 Hz, 1H, Ar-H), 7.58 (t, *J* = 52.5 Hz, 1H, CHF_2_), 7.47–7.41 (m, 2H, Ar-H), 7.31–7.26 (m, 4H, Ar-H), 7.24–7.21 (m, 1H, Ar-H), 6.65 (s, 1H, Ar-H), 6.52 (s, 1H, Ar-H), 3.93–3.91 (m, 4H, morph.), 3.83 (t, *J* = 4.6 Hz, 4H, morph.), 3.68 (s, 2H, CH_2_), 3.44 (s, 2H, CH_2_), 2.50–2.48 (m, 4H, piperaz.), and 2.39–2.37 (m, 4H, piperaz.).^13^C NMR (151 MHz, DMSO-*d*_6_) *δ* 154.9, 150.8, 149.6, 147.0, 144.6, 141.2, 138.2, 134.0, 128.7, 128.1, 126.8, 125.4, 123.9, 120.6, 112.4, 108.5, 95.4, 87.8, 65.5, 62.0, 55.7, 52.6, 52.6, and 48.1.HRMS (ESI/MS): *m*/*z* calculated for C_30_H_32_F_2_N_8_O [M + H]^+^ 558.2667, found 558.2681.

1-[2-({4-[(2-Fluorophenyl)methyl]piperazin-1-yl}methyl)-7-(morpholin-4-yl)pyrazolo[1,5-*a*]pyrimidin-5-yl]-2-(propan-2-yl)-1*H*-benzimidazole (**7**).Compound **7** was prepared from 5-[2-(difluoromethyl)-1*H*-benzimidazol-1-yl]-7-(morpholin-4-yl)pyrazolo[1,5-*a*]pyrimidine-2-carbaldehyde (0.20 g, 0.50 mmol), 1-[(2-fluorophenyl)methyl]piperazine (0.105 mL, 0.12 g, 0.60 mmol) as an amine, DCM (2.0 mL), and sodium triacetoxyborohydride (0.16 g, 0.75 mmol), according to the general procedure for reductive aminationreaction. The resulting crude product was purified by flash chromatography (0–100% AcOEt gradient in heptane) to obtain compound **7** (0.74 g, 0.13 mmol) with a 26% yield.^1^H NMR (600 MHz, DMSO-*d*_6_) *δ* 7.88–7.87 (m, 1H, Ar-H), 7.82 (d, *J* = 7.6 Hz, 1H, Ar-H), 7.57 (t, *J* = 52.5 Hz, 1H, CHF_2_), 7.46–7.41 (m, 2H, Ar-H), 7.32–7.26 (m, 2H, Ar-H), 7.13–7.10 (m, 2H, Ar-H), 6.65 (s, 1H, Ar-H), 6.56 (s, 1H, Ar-H), 4.37 (dd, *J* = 8.4, 6.5 Hz, 1H), 3.93–3.99 (1H), 3.91–3.90 (m, 4H, morph.), 3.86–3.81 (m, 4H, morph.), 3.45 (s, 2H, CH_2_), 3.13 (dd, *J* = 16.2, 8.6 Hz, 1H), 2.99 (dd, *J* = 16.3, 6.4 Hz, 1H), 2.54 (d, *J* = 8.5 Hz, 2H, CH_2_), and 2.40–2.32 (m, 4H, piepraz.).^13^C NMR (151 MHz, DMSO-*d*_6_) *δ* 206.6, 160.7, 155.0, 150.8, 149.4, 147.0, 144.7, 141.2, 134.0, 131.5, 128.9, 125.4, 124.5, 124.0, 123.9, 120.6, 115.0, 112.4, 108.4, 95.2, 87.8, 65.6, 58.2, 54.5, 52.8, 48.1, 43.6, and 30.0.HRMS (ESI/MS): *m*/*z* calculated for C_30_H_31_F_3_N_8_O [M + H]^+^ 576.2564, found 576.2573.

1-[2-({4-[(3-Fluorophenyl)methyl]piperazin-1-yl}methyl)-7-(morpholin-4-yl)pyrazolo[1,5-*a*]pyrimidin-5-yl]-2-(propan-2-yl)-1*H*-benzimidazole (**8**).Compound **10** was prepared from 5-[2-(difluoromethyl)-1*H*-benzimidazol-1-yl]-7-(morpholin-4-yl)pyrazolo[1,5-*a*]pyrimidine-2-carbaldehyde (0.25 g, 0.63 mmol), 1-[(3-fluorophenyl)methyl]piperazine (0.15 g, 0.75 mmol) as an amine, DCM (2.5 mL), and sodium triacetoxyborohydride (0.20 g, 0.94 mmol), according to the general procedure for reductive amination reaction. The resulting crude product was purified by flash chromatography (0–10% MeOH gradient in AcOEt) to obtain compound **8** (0.24 g, 0.41 mmol) with a 65% yield. ^1^H NMR (600 MHz, DMSO-*d*_6_) *δ* 7.84 (dd, *J* = 40.8, 7.5 Hz, 2H, Ar-H), 7.58 (t, *J* = 52.4 Hz, 1H, CHF_2_), 7.47–7.41 (m, 2H, Ar-H), 7.34 (dd, *J* = 14.0, 7.8 Hz, 1H, Ar-H), 7.12–7.03 (m, 3H, Ar-H), 6.65 (s, 1H, Ar-H), 6.52 (d, *J* = 3.5 Hz, 1H, Ar-H), 3.92 (s, 4H, morph.), 3.83 (d, *J* = 4.3 Hz, 4H, morph.), 3.69 (s, 2H, CH_2_), 3.47 (s, 4H, piperaz.), and 2.40 (s, 4H, piperaz.).^13^C NMR (151 MHz, DMSO-*d*_6_) *δ* 163.0, 161.4, 154.9, 150.9, 149.7, 147.0, 141.5, 141.2, 134.0, 130.0, 125.4, 124.6, 123.9, 120.7, 115.0, 113.7, 112.4, 108.5, 95.4, 87.8, 65.5, 61.2, 55.7, 52.6, and 48.2. HRMS (ESI/MS): *m*/*z* calculated for C_30_H_31_F_3_N_8_O [M + H]^+^ 576.2573, found 576.2590.

1-[2-({4-[(4-Fluorophenyl)methyl]piperazin-1-yl}methyl)-7-(morpholin-4-yl)pyrazolo[1,5-*a*]pyrimidin-5-yl]-2-(propan-2-yl)-1*H*-benzimidazole (**9**).Compound **9** was prepared from 5-[2-(difluoromethyl)-1*H*-benzimidazol-1-yl]-7-(morpholin-4-yl)pyrazolo[1,5-*a*]pyrimidine-2-carbaldehyde (0.20 g, 0.50 mmol), 1-[(4-fluorophenyl)methyl]piperazine (0.12 g, 0.60 mmol) as an amine, DCM (2.0 mL), and sodium triacetoxyborohydride (0.16 g, 0.75 mmol), according to the general procedure for reductive amination reaction. The resulting crude product was purified by flash chromatography (0–100% AcOEt gradient in heptane) to obtain compound **9** (0.25 g, 0.44 mmol) with a 88% yield. ^1^H NMR (400 MHz, CDCl_3_) *δ* 7.93–7.91 (m, 1H, Ar-H), 7.66–7.64 (m, 1H, Ar-H), 7.45–7.39 (m, 2H, Ar-H), 7.30–7.26 (m, 3H, Ar-H), 7.16 (s, 1H, Ar-H), 7.01–6.97 (m, 2H, Ar-H), 6.59 (s, 1H, Ar-H), 6.30 (s, 1H, Ar-H), 3.98 (q, *J* = 3.1 Hz, 4H, morph.), 3.92–3.88 (m, 4H, morph.), 3.80 (s, 2H, CH_2_), 3.48 (s, 2H, CH_2_), 2.63 (s, 4H, piperaz.), and 2.51 (s, 4H, piperaz.). ^13^C NMR (101 MHz, CDCl_3_) *δ* 163.2, 160.8, 155.6, 151.3, 150.1, 147.5, 144.7, 141.8, 134.6, 130.7, 130.6, 125.7, 124.2, 121.6, 115.1, 114.9, 111.7, 111.7, 109.3, 96.6, 87.3, 66.2, 62.2, 56.4, 53.1, 48.5, 31.6, 22.6, and 14.1.HRMS (ESI/MS): *m*/*z* calculated for C_30_H_31_F_3_N_8_O [M + H]^+^ 576.2573, found 576.2588.

#### 3.3.3. General Procedure for the Amidation Reaction

The corresponding amine (1.05 eq), 2-(7-aza-1*H*-benzotriazole-1-yl)-1,1,3,3-tetramethyluronium hexafluorophosphate (HATU) (1.1 eq), and triethylamine (1.5 eq) were added to the solution of substituted 5-chloro-pyrazolo[1,5-*a*]pyrimidine derivative (1.0 eq) in solvent (10 mL/1 g pyrazolo[1,5-*a*]pyrimidine derivative). The reaction mixture was stirred at room temperature for 2 h. Then, water was added to the mixture and the phases were separated. The aqueous phase was extracted three times with a solvent, while combined organic phases were dried over anhydrous sodium sulfate, filtered, and concentrated. The resulting residue was purified by flash chromatography.

1-[2-(4-Benzylpiperazine-1-carbonyl)-7-(morpholin-4-yl)pyrazolo[1,5-*a*]pyrimidin-5-yl]-2-(difluoromethyl)-1*H*-benzimidazole (**10**).Compound **10** was prepared from 5-[2-(difluoromethyl)-1*H*-benzimidazol-1-yl]-7-(morpholin-4-yl)pyrazolo[1,5-*a*]pyrimidine-2-carboxylic acid (0.45 g, 1.09 mmol), 1-benzylpiperazine (0.20 g, 1.14 mmol), HATU (0.46 g, 1.19 mmol), TEA (0.23 mL, 0.16 g, and 1.63 mmol), and DCM (4.0 mL), according to the general procedure for amidation reaction. The resulting crude product was purified by flash chromatography (0–10% MeOH gradient in AcOEt) to obtain compound **10** (0.32 g, 0.56 mmol) as a light yellow solid with a 51% yield.^1^H NMR (600 MHz, DMSO-*d*_6_) *δ* 7.89 (d, *J* = 7.6 Hz, 1H, Ar-H), 7.82 (d, *J* = 7.8 Hz, 1H, Ar-H), 7.58 (t, *J* = 52.4 Hz, 1H, CHF_2_), 7.48–7.42 (m, 2H, Ar-H), 7.35–7.31 (m, 4H, Ar-H), 7.26 (td, *J* = 5.9, 2.6 Hz, 1H, Ar-H), 6.83 (s, 1H, Ar-H), 6.81 (s, 1H, Ar-H), 3.91–3.90 (m, 4H, morph.), 3.83 (t, *J* = 4.5 Hz, 4H, morph.), 3.72 (d, *J* = 41.1 Hz, 4H, piperaz.), 3.53 (s, 2H, CH_2_), and 2.44 (dt, *J* = 23.1, 4.6 Hz, 4H, piperaz.).^13^C NMR (151 MHz, DMSO-*d*_6_) *δ* 161.6, 151.1, 150.3, 149.2, 147.7, 144.6, 141.2, 137.7, 134.0, 128.9, 127.0, 125.5, 124.0, 120.7, 108.5, 65.5, 52.9, 48.4, 30.9, and 26.8. HRMS (ESI/MS): *m*/*z* calculated for C_30_H_30_F_2_N_8_O_2_ [M + H]^+^ 572.2460, found 572.2477.

2-(Difluoromethyl)-1-(2-{4-[(2-fluorophenyl)methyl]piperazine-1-carbonyl}-7-(morpholin-4-yl)pyrazolo[1,5-*a*]pyrimidin-5-yl)-1*H*-benzimidazole (**11**).Compound **11** was prepared from 5-[2-(difluoromethyl)-1*H*-benzimidazol-1-yl]-7-(morpholin-4-yl)pyrazolo[1,5-*a*]pyrimidine-2-carboxylic acid (0.20 g, 0.48 mmol), 1-(2-fluorobenzyl)piperazine (0.89 mL, 0.10 g, and 0.51 mmol), HATU (0.20 g, 0.53 mmol), TEA (0.10 mL, 0.073 g, and 0.72 mmol), and DCM (2.0 mL), according to the general procedure for amidationreaction. The resulting crude product was purified by flash chromatography (0–5% MeOH gradient in AcOEt) to obtain compound **11** (0.17 g, 0.29 mmol) as a white solid with a 60% yield. ^1^H NMR (400 MHz, DMSO-*d*_6_) *δ* 7.89 (d, *J* = 7.3 Hz, 1H, Ar-H), 7.82 (d, *J* = 7.3 Hz, 1H, Ar-H), 7.61 (t, *J* = 52.6 Hz, 1H, CHF_2_), 7.47–7.41 (m, 3H, Ar-H), 7.33 (d, *J* = 7.7 Hz, 1H, Ar-H), 7.20–7.15 (m, 2H, Ar-H), 6.83 (s, 1H, Ar-H), 6.80 (s, 1H, Ar-H), 3.83 (d, *J* = 3.8 Hz, 4H, morph.), 3.72 (d, *J* = 27.4 Hz, 4H, morph.), 3.60 (s, 2H,), and 2.49–2.46 (m, 4H).^13^C NMR (101 MHz, DMSO-*d*_6_) *δ* 161.7, 159.6, 151.2, 150.2, 149.2, 147.8, 144.7, 141.2, 134.0, 131.7, 129.2, 125.5, 124.2, 124.0, 120.7, 115.3, 112.4, 108.5, 96.7, 89.3, 65.6, 54.4, 52.7, 52.1, 48.4, and 41.8.HRMS (ESI/MS): *m*/*z* calculated for C_30_H_29_F_3_N_8_O_2_ [M + H]^+^ 590.2366, found 590.2381.

2-(Difluoromethyl)-1-(2-{4-[(3-fluorophenyl)methyl]piperazine-1-carbonyl}-7-(morpholin-4-yl)pyrazolo[1,5-*a*]pyrimidin-5-yl)-1*H*-benzimidazole (**12**).Compound **12** was prepared from 5-[2-(difluoromethyl)-1*H*-benzimidazol-1-yl]-7-(morpholin-4-yl)pyrazolo[1,5-*a*]pyrimidine-2-carboxylic acid (0.24 g, 0.59 mmol), 1-(3-fluorobenzyl)piperazine (0.12 g, 0.62 mmol), HATU (0.25 g, 0.65 mmol), TEA (0.12 mL, 0.09 g, and 0.88 mmol), and DCM (2.0 mL), according to the general procedure for amidation reaction. The resulting crude product was purified by flash chromatography (0–15% MeOH gradient in AcOEt) to obtain compound **12** (0.13 g, 0.21 mmol) as a white solid with a 36% yield.^1^H NMR (600 MHz, DMSO-*d*_6_) *δ* 7.89 (d, *J* = 7.6 Hz, 1H, Ar-H), 7.82 (d, *J* = 8.1 Hz, 1H, Ar-H), 7.58 (t, *J* = 52.4 Hz, 1H, CHF_2_), 7.48–7.42 (m, 2H, Ar-H), 7.37 (dd, *J* = 14.1, 7.9 Hz, 1H, Ar-H), 7.17–7.15 (m, 2H, Ar-H), 7.10–7.07 (m, 1H, Ar-H), 6.84 (s, 1H, Ar-H), 6.81 (s, 1H, Ar-H), 3.91–3.90 (m, 4H, morph.), 3.84–3.82 (m, 4H, morph.), 3.73 (d, *J* = 41.5 Hz, 4H, piperaz.), 3.56 (s, 2H), 2.68 (s, 3H), 2.44 (d, *J* = 4.5 Hz, 4H, piperaz.), and 2.43 (t, *J* = 4.5 Hz, 2H).^13^C NMR (151 MHz, DMSO-*d*_6_) *δ* 162.2, 161.7, 151.1, 150.3, 149.2, 147.7, 144.6, 141.2, 140.9, 134.0, 130.1, 125.5, 124.7, 124.0, 120.7, 115.2, 113.8, 112.4, 108.5, 96.7, 89.2, 65.5, 48.4, and 38.2.HRMS (ESI/MS): *m*/*z* calculated for C_30_H_29_F_3_N_8_O_2_ [M + H]^+^ 590.2366, found 590.2385.

2-(Difluoromethyl)-1-(2-{4-[(4-fluorophenyl)methyl]piperazine-1-carbonyl}-7-(morpholin-4-yl)pyrazolo[1,5-*a*]pyrimidin-5-yl)-1*H*-benzimidazole (**13**).Compound **13** was prepared from 5-[2-(difluoromethyl)-1*H*-benzimidazol-1-yl]-7-(morpholin-4-yl)pyrazolo[1,5-*a*]pyrimidine-2-carboxylic acid (0.15 g, 0.36 mmol), 1-(4-fluorobenzyl)piperazine (0.075 g, 0.38 mmol), HATU (0.15 g, 0.40 mmol), TEA (0.076 mL, 0.05 g, and 0.54 mmol), and DCM (1.5 mL), according to the general procedure for amidationreaction. The resulting crude product was purified by flash chromatography (0–5% MeOH gradient in AcOEt) to obtain compound **13** (0.13 g, 0.22 mmol) as a white solid with a 62% yield. ^1^H NMR (600 MHz, DMSO-*d*_6_) *δ* 7.89 (d, *J* = 7.4 Hz, 1H, Ar-H), 7.82 (d, *J* = 7.8 Hz, 1H, Ar-H), 7.58 (t, *J* = 52.6 Hz, 1H, CHF_2_), 7.48–7.42 (m, 2H, Ar-H), 7.37–7.35 (m, 2H, Ar-H), 7.15 (t, *J* = 8.7 Hz, 2H, Ar-H), 6.83 (s, 1H, Ar-H), 6.81 (s, 1H, Ar-H), 3.91 (t, *J* = 4.6 Hz, 4H, morph.), 3.83 (t, *J* = 4.6 Hz, 4H, morph.), 3.72 (d, *J* = 42.1 Hz, 4H, piperaz.), 3.51 (s, 2H, CH_2_), and 2.43 (d, *J* = 23.5 Hz, 3H).^13^C NMR (151 MHz, DMSO-*d*_6_) *δ* 161.7, 151.1, 150.3, 149.2, 147.8, 144.7, 141.2, 134.0, 130.7, 130.7, 125.5, 124.0, 120.7, 115.0, 114.8, 112.4, 108.5, 96.7, 89.3, 65.5, 60.8, and 48.4.HRMS (ESI/MS): *m*/*z* calculated for C_30_H_29_F_3_N_8_O_2_ [M + H]^+^ 590.2366, found 590.2380.

### 3.4. In Vitro PI3Kδ Inhibition Assay

All compounds were tested by a biochemical assay that measured the inhibition of phosphatidylinositol (4,5)-bisphosphate (PIP2) production by PI3K isoform. The potency of the tested compounds was assessed by determining the ability of PI3K*δ* enzymes (Merck Millipore) to convert ATP to ADP during an enzymatic reaction in the presence of these compounds at decreasing doses. The experiments were carried out using the ADP-Glo kinase assay kit (Promega), according to the manufacturer’s protocol. PIP2 lipid vesicles containing phosphoserine (ThermoFisher Scientific, Waltham, MA, USA) were used as a substrate in the enzymatic reaction. The composition of the reaction mixture and reaction conditions for PI3K*δ* were as follows: concentration of PI3K*δ* enzyme: 10 ng; reaction temperature and time: 25 °C and 1 h; final concentration of PIP2 substrate: 30 μM; and reaction buffer: 50 mM of HEPES (pH 7.5), 50 mM NaCl, 3 mM MgCl_2_, and 0.025 mg/mL of BSA.

After the reaction, the ADP-Glo reagent and the kinase detection reagent were sequentially added. The reaction mixture was incubated for 40 min (25 °C, 600 rpm) after the addition of each reagent. Finally, luminescence intensity was measured and the IC_50_ value was calculated using GraphPad Prism 7 software (GraphPad, Boston, MA, USA). The results were presented as the mean value of IC_50_ obtained from at least two independent experiments.

### 3.5. Computational Workflow Used to Predict the Most Potent Fluorine Derivative

We used a previously described computational workflow involving IFD, MD simulations, and QPLD combined with energy calculations (applying the MM-GBSA method). The crystal structure of PI3K*δ* protein (PDB ID: 2WXL) [25] that was successfully used in our previous study to support the SAR analysis [19] was retrieved from Protein Data Bank [26,27].

#### 3.5.1. IFD

The three-dimensional structures of the ligands were prepared using LigPrep v3.6 [28], and appropriate ionization states at pH 7.0 ± 0.5 were assigned using Epik v3.4 [29]. The Protein Preparation Wizard tool [28] was used to assign bond orders and appropriate amino acid ionization states and to check for steric clashes for the PI3K*δ* crystal structure. The receptor grid was generated (OPLS3 force field [30]) by centering the grid box with a size of 8 Å on crystalized ligands (ZSTK474). Automated flexible docking [31,32] of nonfluorinated compounds was performed using Glide v6.9 [33,34,35] at the SP level.

#### 3.5.2. MD

MD simulations (100 ns long) were performed using Schrödinger Desmond software [36,37]. Each ligand–receptor complex selected based on the IFD analysis was joined with the POPC (309.5 K) membrane bilayer. The system was solvated by water molecules described by the TIP4P potential [38] in orthorhombic box with a distance of 10Å from the complex, using the OPLS3e force field [30] for all atoms. NaCl (0.15 M) was added to mimic the ionic strength inside the cell. The simulations were carried out using the NPAT protocol at a temperature of 309.5 K and a pressure of 1013.25 hPa.

#### 3.5.3. QPLD

The grids for the receptors were generated (OPLS3e force field) by centering the grid box (size 8 Å) on a ligand. Docking of all fluorinated compounds was performed by the QPLD [16] procedure involving the QM-derived ligand atomic charges in the protein environment at the BLYP/cc-pVDZ level [39,40]. Five poses were obtained for each ligand.

#### 3.5.4. Binding-Free Energy Calculations

Using GBSA, the binding-free energy was calculated based on the ligand–receptor complexes generated by the QPLD procedure. The ligand poses were minimized using the local optimization feature in Prime. The distance of the flexible residue from a ligand pose was set to 6 Å. Ligand charges obtained in the QPLD stage were used. The energies of complexes were calculated with the OPLS3e force field and the GBSA continuum solvent model. To assess the influence of a given substituent on binding, ΔΔG was calculated as the difference between the binding-free energy (ΔG) of the nonfluorinated compound and its fluorinated analogs.

## 4. Conclusions

We evaluated a workflow involving IFD, MD, and QM/MM docking (QPLD) with the MM-GBSA calculation to score halogenated derivatives of CPL302415, a clinical PI3K*δ* selective inhibitor, and compared the results with those obtained by the Glide scoring function. Additionally, we synthesized a series of novel fluorinated compounds to estimate the accuracy of the pose prediction in the molecular docking procedure. 

We found that a properly prepared catalytic (binding) pocket for chemical cores in the IFD and MD stages, combined with QM-derived atomic charges, can be effectively used for prediction, as well as to improve the discrimination of active compounds from inactive ones. Moreover, the standard approach (rigid docking) seems to be less effective at scoring halogenated derivatives due to the fixed atomic charges and the fact that it does not consider the response and indictive effects caused by fluorine. Our results suggest that the proposed computational workflow may be a valuable tool for the rational design of new halogenated drugs.

## Figures and Tables

**Figure 1 molecules-28-03531-f001:**
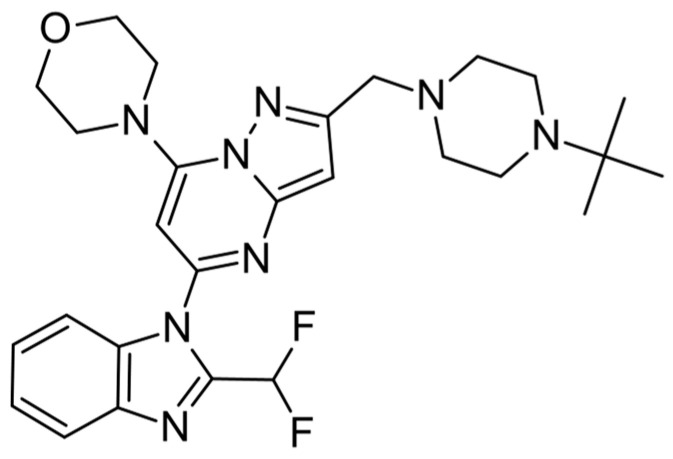
Structure of CPL302415.

**Figure 2 molecules-28-03531-f002:**
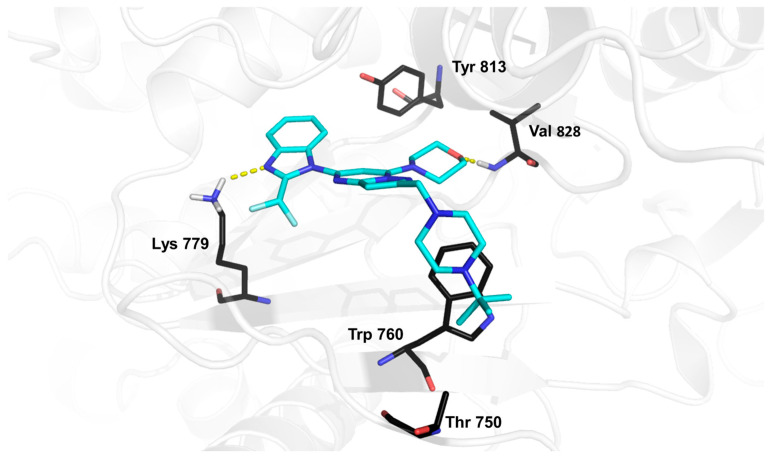
Illustration of the binding mode of CPL302415 (**2**) in the catalytic center of PI3K*δ* obtained by molecular docking [19].

**Figure 3 molecules-28-03531-f003:**
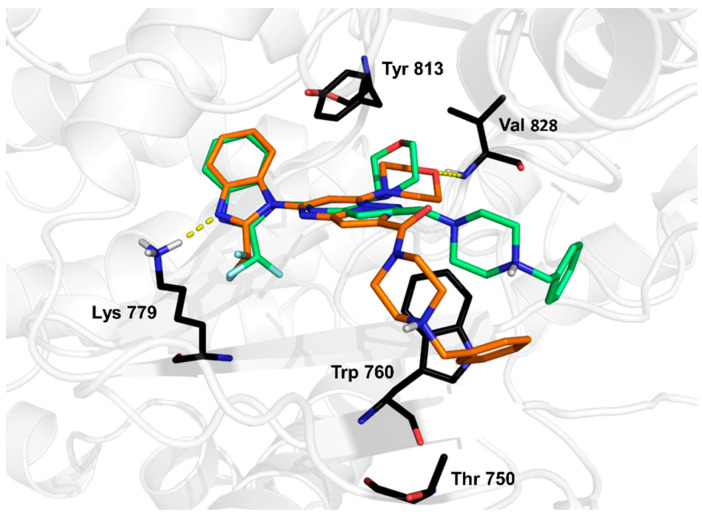
Illustration of the binding modes of compounds **6** (green) and **10** (orange) in the catalytic center of PI3K*δ* obtained by molecular docking.

**Figure 4 molecules-28-03531-f004:**
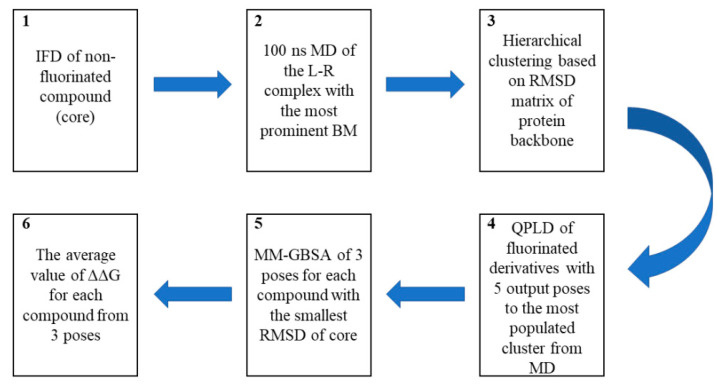
Computational workflow used to predict the most potent fluorine derivative. The workflow began with the IFD of a nonfluorinated compound (core) (**1**) followed by 100 ns-long MD simulations (**2**), which were then clustered based on the RMSD matrix of the protein backbone (**3**). All derivatives were docked to the three most frequently observed conformations of the protein using the QPLD algorithm (**4**). Using the MM-GBSA approach, the binding energy (ΔG) was calculated for the three conformations of the ligand with the smallest RMSD of the core to nonfluorinated compounds (**5**). Finally, the difference in the interaction energy between the most active compound and subsequent isomers (ΔΔG) was calculated (**6**).

**Figure 5 molecules-28-03531-f005:**
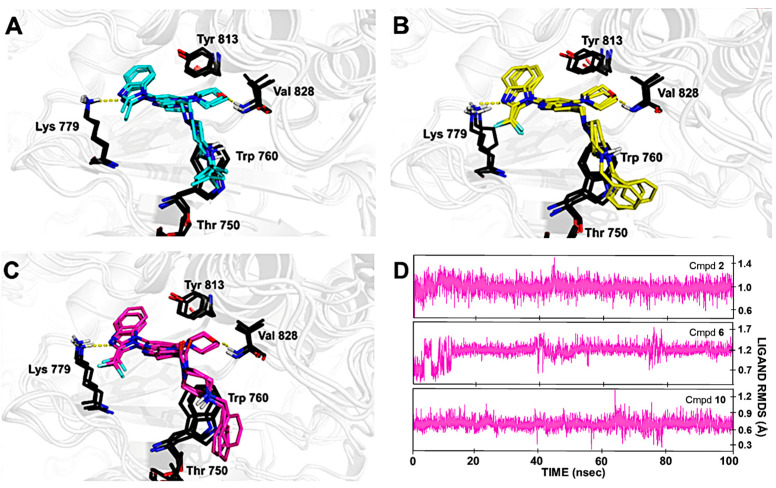
Superposition of the binding modes of core (**A**) *tert*-butyl piperazine (cyan), (**B**) benzylpiperazine (yellow), and (**C**) 4-benzylpiperazine-1-carbonyl (magenta) in the PI3K*δ* catalytic sites. The selected complexes were the most populated conformations resulting from the clustering of the MD trajectories. The RMSD (Å) of compounds **2**, **6**, and **10**, respectively, during MD simulations (**D**) indicates the stability of the ligand with respect to the protein and its binding pocket. Ligand RMSD is a measure of the internal fluctuations of ligand atoms.

**Figure 6 molecules-28-03531-f006:**
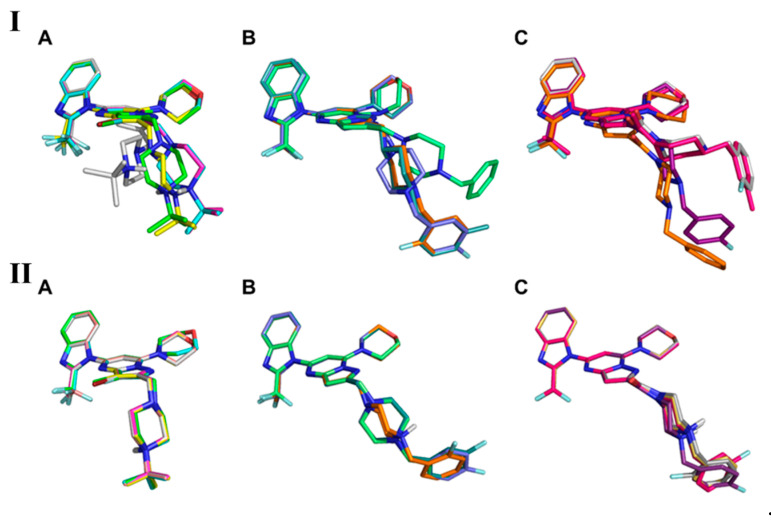
Comparison of the binding mode coherence of core (A) *tert*-butyl piperazine, (B) benzylpiperazine, and (C) 4-benzylpiperazine-1-carbonyl obtained using the standard (rigid) molecular docking approach (**I**) and the proposed in silico workflow (**II**).

**Table 1 molecules-28-03531-t001:** Influence of the fluorine atom(s) on PI3K*δ* inhibition and the respective docking scores for each derivative.

** 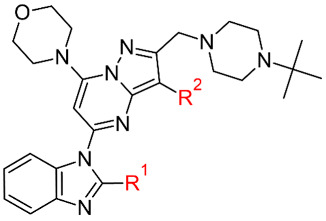 **					
**Compound**	**R^1^**	**R^2^**	**IC_50_ PI3K*δ* (nM) ^a^**	**Docking Score**	
				**First Pose**	**Average of Top Three Poses**
**1**	CH_3_	H	236	−9.3	−8.8
**2 (CPL302415)**	CHF_2_	H	18	−9.9	−9.8
**3**	CF_3_	H	907	−9.6	−9.3
**4**	CHF_2_	Cl	44	−10.4	−10.1
**5**	CHF_2_	Br	50	−10.5	−10.1

^a^ IC_50_ values were determined as the mean from two independent experiments.

**Table 2 molecules-28-03531-t002:** Influence of the fluorine atom(s) on PI3K*δ* inhibition and the respective docking scores for each compound.

** 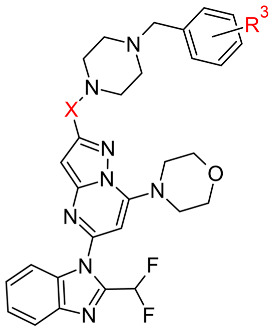 **					
**Compound**	**X**	**R^3^**	**IC_50_ PI3K*δ* (nM) ^a^**	**Docking score**	
				**First Pose**	**Average of Top Three Poses**
**6**	CH_2_	-	118	−6.8	−6.4
**7**	CH_2_	*o*-F	640	−10.8	−10.4
**8**	CH_2_	*m*-F	751	−9.0	−8.3
**9**	CH_2_	*p*-F	489	−9.2	−8.1
**10**	CO	-	275	−10.1	−9.1
**11**	CO	*o*-F	212	−10.3	−9.3
**12**	CO	*m*-F	92	−10.9	−10.8
**13**	CO	*p*-F	181	−10.5	−10.1

^a^ IC_50_ values were determined as the mean from two independent experiments.

**Table 3 molecules-28-03531-t003:** Influence of the fluorine atom(s) on PI3K*δ* inhibition and the respective ΔG and ΔΔG ^a^ scores for each compound.

Compound	R^1^	R^2^	IC_50_ PI3K*δ* (nM) ^b^	$\overset{-}{\Delta G}$	ΔΔG (kcal/mol)
**1**	CH_3_	H	236	−75.0	–
**2 (CPL302415)**	CHF_2_	H	18	−82.6	−7.6
**3**	CF_3_	H	907	−69.6	5.4
**4**	CHF_2_	Cl	44	−81.3	1.3 ^c^
**5**	CHF_2_	Br	50	−81.2	1.4 ^c^

^a^ The interaction energy gain averaged by three ligand–receptor complexes of each derivative selected from the MD simulations. ^b^ IC_50_ values were determined as the mean from two independent experiments. ^c^ ΔΔG values calculated as a difference between a given derivative and its nonhalogenated (difluoromethyl)-1*H*-benzimidazole analog.

**Table 4 molecules-28-03531-t004:** Influence of the fluorine atom(s) on PI3K*δ* inhibition and the respective ΔG and ΔΔG ^a^ scores for each compound.

Compound	X	R^3^	IC_50_ PI3K*δ* (nM) ^b^	$\overset{-}{\Delta G}$	ΔΔG (kcal/mol)
**6**	CH_2_	-	118	−86.7	–
**7**	CH_2_	*o*-F	640	−83.4	3.3
**8**	CH_2_	*m*-F	751	−81.8	4.9
**9**	CH_2_	*p*-F	489	−84.1	2.6
**10**	CO	-	275	−78.9	–
**11**	CO	*o*-F	212	−83.8	−4.9
**12**	CO	*m*-F	92	−86.6	−7.7
**13**	CO	*p*-F	181	−83.8	−4.9

^a^ The interaction energy gain averaged by three ligand–receptor complexes of each derivative selected from the MD simulations. ^b^ IC_50_ values were determined as the mean from two independent experiments.

## Data Availability

Data are contained within the article and Appendix A.

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
