# Peer review of "Tuning the Biological Activity of PI3Kδ Inhibitor by the Introduction of a Fluorine Atom Using the Computational Workflow"

_molecules, 2023, doi:10.3390/molecules28083531_

Round 1

Reviewer 1 Report

The authors evaluated a workflow involving IFD, MD, and QM/MM docking (QPLD) with the MM-GBSA calculation to score halogenated derivatives of a clinical PI3Kδ 546 selective inhibitor, and compared the results with those obtained by Glide scoring function, this computational workflow is a valuable tool for the rational designing of new halogenated drugs. I would recommend accepting this research for publication.

Author Response

The authors evaluated a workflow involving IFD, MD, and QM/MM docking (QPLD) with the MM-GBSA calculation to score halogenated derivatives of a clinical PI3Kδ 546 selective inhibitor, and compared the results with those obtained by Glide scoring function, this computational workflow is a valuable tool for the rational designing of new halogenated drugs. I would recommend accepting this research for publication.

We would like to thank the reviewer for the time spent reading our manuscript and for the kind review.

Reviewer 2 Report

The authors have carried out an investigation on tuning the biological activity of PI3Kδ inhibitor by the introduction of a fluorine atom using the computational workflow. Overall, the manuscript submitted by Wojciech et al. is an interesting study and describes exciting findings. In the reviewer’s opinion, the manuscript is properly organized and referenced. The reviewer thinks that a potential reader can understand the details by consulting the main data outlined in the current version of the manuscript with a few exceptions underlined below. The abstract adequately conveys the whole content of the article, and the conclusions are appropriate in the reviewer’s understanding. Although there are many typos and linguistic errors that need the author’s attention however, the overall write-up is clear and to the point, which is crucial for a research article. Herein, I would like to recommend the current version of this manuscript for publication in Molecules but after major revision. To this end, there are some recommendations and suggestions related to the manuscript and authors should take into account these suggestions before submitting a revised version.

1)       The authors did decent efforts to come up with an outcome. Why not the authors performed in vivo studies for the synthesized analogs? Only in silico execution does not provide a sufficient basis for the potency of any analogs for further development. IC50 values should be determined in vivo models to provide coherence between the two approaches (computational and experimental). Reviewer feels that current biological activity values are nothing more than theoretical calculations. In the reviewer’s opinion synthesized analogs should be tested in vivo first and then incorporate in vivo results to have a good comparison to benefit the readership of the journal.

2)       What are the advantages of using CPL302415 as a core structure, except it is a promising preclinical candidate, compared with other PI3Kδ inhibitors developed clinically so far? Why did the author select CPL302415 for the current study although numerous similar analogs of CPL302415 are known?

3)       Reviewer feels a random selection of substitution patterns during the work plan is not adequate to come up with a decent conclusion/potent molecule. Please justify the selection around the core.

4)       The authors are suggested to carefully check the existence of 6-13 using a SciFinder and the report should be attached during the submission of the revised submission.

5)       Table 4. ortho, meta etc., should be given in italics.

6)       Table 2. R2 is missing. Rectify this error.

7)       Mention the compound numbers that which author synthesized, and discussed in this work (Line: 229).

8)       Line: 230. Describe a specific scheme used for the synthesis of analogs and upload in the SI file.

9)       Section (3.2 General information) should be moved to the SI file to concise the article.

10)   Lines 99-102 are not clear. Add a synthetic scheme to make it clearer.

11)   HNMR for compound 3 shows impurity. Even CNMR is also showing the impurity. Replace it with a reasonable version.

12)   The authors stated that compounds 1 and 2 were synthesized according to the procedures described in previous publications (19, 23). It is recommended to either mention one reference or specify which ref was used for which purpose.  

Author Response

Our answers are attached.

Reviewer 3 Report

 At the beginning of paragraph 2 the bad correlation of the docking score with the IC50 values are discussed. It is not surprising, since scoring functions were not developed for ranking active compounds, but rather for the discrimination of active and inactive compounds. Actually, the “docking score” worked extremely well by identifying the top compound 2 and 12 in Tables 1,2, and 3. Even in Table 4 compound 7 is very close to the top compound 13. Calculating correlation between docking score and IC50 values for such a low number of active ligands is a bit misleading in this case, however understandable in the light of the “take home” message of the manuscript.

The applied protocol incorporates a high level of the binding site flexibility into account, and uses precise atomic charges obtained from quantum chemical calculations. The protocol makes perfectly sense, since there are difficult targets for structure based drug design, where incorporation of the binding site flexibility may be extremely important. For fluorine derivatives with high electronegativity the usage of precise atomic charges can be crucial. However it should be demonstrated, what is the accuracy gain of these approaches in the case of the PI3Kd lipid kinase target. It would be interesting to see how the MM-GBSA methods alone, which already incorporates a basic level of receptor flexibility, would have performed on the investigated compounds? Especially with the usage of precise quantum mechanics derived atomic charges.

The investigated PI3Kd lipid kinase is not a transmembrane, but a membrane associated protein. Accordig to line 519 “ligand–receptor complex selected for IFD was immersed into a POPC membrane bilayer”. The word immersion is a bit misleading in this case, please include a picture of the whole system (without the water molecules) into the Supplement. Details about the positioning of the protein on the membrane should also be added to the manuscript. Information about the molecular dynamics simulation should be added to the Materials and Methods section (size of the prepared system, simulation protocol, temperature, pressure,. …). Next to the ligand RMSD curves, additional information of the simulation could be added to the Supplement.

The effect of the MD simulations on the MM-GBSA DG(bind) values and the binding poses and the should be demonstrated in the manuscript in from of a figure, and by giving ligand RMSD values using both the standard Glide and the IFD poses as references.

There are a couple of suggestion regarding the figures in the manuscript:

Figure 1 and the molecular structures in Tables 1 and 2 should be harmonized, showing the same orientation. Merging all molecular structure to Figure 1 A,B, and C could also be considered.

In Table 2 an R1 group is marked on the molecule, but R2 is listed in the column header. Since these groups are different from the R2 groups in Table 1, the notation R3 would be favored in both Table 2 and 4.

In the legend of Figure 3 the different compound should be identified by their colors.

The Thr750 and Tyr813 residues present on Figure 3 could be added also to Figure 2, to provide a consistent view of the ligand binding pocket.

The hydrogen bond between compound 10 and the Val828 residue is hardly recognizable on Figure 3, it should be changed.

On Figure 5 the RMS plots should get a notation like Fig 5D. On the printed version of the same figure the “C” letter was truncated.

In the legend of Figure 6, the (II) poses were obtained probably by the “standard” rigid Glide rigid receptor docking procedure. If the top3 poses are pictured, it should be included in the figure legend. It would be interesting to see how the poses obtained by the Induced Fit Docking procedure compare to them without the MD simulations.

There were “Bookmark not defined errors in the Supplementary”.

Citation of the used programs (reference 34) is inappropriate. Citation for Desmond and possibly IFD are missing.

Author Response

Our responses are attached.

Round 2

Reviewer 2 Report

Dear Editor

The authors have performed the necessary modifications in response to the queries. The authors have further expanded the manuscript content with newer pieces of information where necessary/required. I would like to recommend the publication of this study in its current version.

Author Response

We want to thank the reviewer for the suggestions and beneficial comments which we believed to improved the manuscript.

Reviewer 3 Report

 Most of the remarks were taken into account, except that regarding Figure 2, and the major one about demonstrating the power of the workflow.

 I was thinking about a minor revision, but I think that the manuscript would have a higher value by analyzing the effect of the flexibility and quantum calculations present in the workflow separately, rather than simply presenting results of the protocol. Since the most time consuming  MD calculations were the already performed, only about a handful of 5/90 minute long calculations should be performed. 

Author Response

We performed the calculations suggested by the Reviewer. QPLD protocol which uses MM-GBSA does not contain any flexibility itself. This approach is based on the QM-based charges used then in standard (rigid) molecular docking. The results were integrated with the manuscript and two new tables were added to the supplementary information.

Round 3

Reviewer 3 Report

 Thank you for your effort by providing additional MM-GBSA calculations! I agree that MM-GBSA is not a method dealing explicitly with protein flexibility, but during the minimizations, residues in the proximity of the ligand are allowed to move and adopt to the ligand structure, as you also state in line 557 (with the usage of the "flexible residue" term). In my view this is a minimal implementation of protein flexiblity. My intention was to convince you to incorporate additional results, which would make your manuscript a good example for demonstrating the effect of protein flexibility incorporated at different levels. You still could do it by performing new calculations on the IFD produced poses (before MD), and could devote a couple of sentences for their interpretation.

 My previous comment regarding figure 2 was not answered. According your first answer "Figure 2 came from the original publication [19]", so it was not modified according the suggestions. In reference 19 the only figure representing a protein structure is Figure 2. Although Figure 2 of the current manuscript shows high similarity with Figure 2A of reference 19, the latter one includes 6 side chains, while the former one only 3. Since Figure 2 is obviously not taken from reference 19, probably only the docking pose, it could be easily modified to contain the relevant residues for the other compounds, presented in Figure 3. I do not insist on changing Figure 2, but your first answer was not acceptable, and there was no second answer.

 The comments regarding table 2 and 4 were not answered. In Table 2 compounds are numbered from 6 to 13, while in Table 4 numbering is shifted from 7 to 14.  There is no compound 6 in Table 3 or 4. Was this your intention? For compound 6 in Table 2 and compound 7 in Table 4 the IC50 values are equal, but in the second column (X) the groups are different (CH2 vs H). Please clarify!

"Bookmark not defined." error is still present in the supplement.

Both "induced-fit" and "induced fit" spelling is still used in the text.

Author Response

We would like to thank the Reviewer for the time spent reading our manuscript and for the kind review, as well as all comments and guidance, which made our manuscript better and more relevant.

Thank you for your effort by providing additional MM-GBSA calculations! I agree that MM-GBSA is not a method dealing explicitly with protein flexibility, but during the minimizations, residues in the proximity of the ligand are allowed to move and adopt to the ligand structure, as you also state in line 557 (with the usage of the "flexible residue" term). In my view this is a minimal implementation of protein flexiblity. My intention was to convince you to incorporate additional results, which would make your manuscript a good example for demonstrating the effect of protein flexibility incorporated at different levels. You still could do it by performing new calculations on the IFD produced poses (before MD), and could devote a couple of sentences for their interpretation.

We would like to thank the reviewer for convincing us to perform additional calculations, the results of which not only demonstrated the utility of our workflow but additionally made us realize how important the flexibility of the binding pocket is in molecular modeling studies.

 My previous comment regarding figure 2 was not answered. According to your first answer "Figure 2 came from the original publication [19]", so it was not modified according the suggestions. In reference 19 the only figure representing a protein structure is Figure 2. Although Figure 2 of the current manuscript shows high similarity with Figure 2A of reference 19, the latter one includes 6 side chains, while the former one only 3. Since Figure 2 is obviously not taken from reference 19, probably only the docking pose, it could be easily modified to contain the relevant residues for the other compounds, presented in Figure 3. I do not insist on changing Figure 2, but your first answer was not acceptable, and there was no second answer.

We agree with the reviewer and, of course, Figure 2 has been revised as recommended. It has been changed in the current version of the manuscript.

 The comments regarding table 2 and 4 were not answered. In Table 2 compounds are numbered from 6 to 13, while in Table 4 numbering is shifted from 7 to 14.  There is no compound 6 in Table 3 or 4. Was this your intention? For compound 6 in Table 2 and compound 7 in Table 4 the IC50 values are equal, but in the second column (X) the groups are different (CH2 vs H). Please clarify!

Thank you very much for pointing out such an obvious mistake. This has been corrected in the current version of the manuscript.

"Bookmark not defined." error is still present in the supplement.

We have checked all the links in the supplements several times and hope that in the Reviewer's version, they will all work.

Both "induced-fit" and "induced fit" spelling is still used in the text.

Thank you for pointing out this inconsistency. It has been corrected in the current version of the manuscript.